# Similar functional composition of fish assemblages despite contrasting levels of habitat degradation on shallow Caribbean coral reefs

**Manuel Olán-González**[1,2], **Patricia Briones-Fourzán**[3], **Enrique Lozano-Álvarez**[3], **Gilberto Acosta-González**[4], **Lorenzo Alvarez-Filip**[2]*

**1** Posgrado en Ciencias del Mar y Limnología, Universidad Nacional Autónoma de México, Coyoacán, Ciudad de México, México, **2** Biodiversity and Reef Conservation Laboratory. Unidad Académica de Sistemas Arrecifales, Instituto de Ciencias del Mar y Limnología, Universidad Nacional Autónoma de México, Puerto Morelos, Quintana Roo, México, **3** Laboratorio de Ecología de Crustáceos. Unidad Académica de Sistemas Arrecifales, Instituto de Ciencias del Mar y Limnología, Universidad Nacional Autónoma de México, Puerto Morelos, Quintana Roo, México, **4** Centro de Investigación Científica de Yucatán A.C. Unidad de Ciencias del Agua, Cancún, Quintana Roo, México

* lorenzo@cmarl.unam.mx

**Data Availability Statement:** All relevant data are within the paper and its Supporting Information files.

## Abstract

Functional trait-based approaches provide an opportunity to assess how changes in habitat affect the structure of associated communities. Global analyses have found a similarity in the composition of reef fish functional traits despite differences in species richness, environmental regimes, and habitat components. These large-scale patterns raised the question of whether this same stability can be observed at smaller spatial scales. Here, we compared the fish trait composition and their functional diversity in two Caribbean shallow coral reefs with contrasting levels of habitat degradation: Limones (>30% cover), constituted mainly by colonies of *Acropora palmata* and Bonanza, a reef with extensive areas of dead *Acropora* structures, dominated by algae. To characterize the functional structure of fishes on each reef, we calculated the community-weighted mean trait values (CWM), functional richness, functional evenness, functional dispersion, and functional originality. Despite the differences in habitat quality, reefs exhibited a similar proportion and common structure on fish functional traits. Functional richness and functional evenness differed significantly, but functional dispersion and functional originality did not show differences between reefs. The greater niche complexity driven by the high availability of microhabitats provided by *A. palmata* may explain the higher functional richness in Limones, whereas the reef degradation in Bonanza may contribute to a higher functional evenness because of a similar distribution of abundance per fish trait combinations. Our results suggest that widespread degradation on Caribbean reefs has limited the type, variety, and range of traits, which could lead to a functional homogenization of fish communities even at local scales.

**Funding:** The data collection of this research was supported by Project UNAM-DGAPA-PAPIIT, IN-205614 granted to E.L.-A. This work was also supported by a Doctoral scholarship CONACYT, 707432 granted to M.O.-G.

**Competing interests:** The authors have declared that no competing interests exist.

## Introduction

Habitat loss and degradation are leading drivers of declining biodiversity and alteration of ecosystem processes and functioning [1, 2]. Trait-based functional approaches are rapidly gaining attention as a tool to evaluate how these changes in habitat quality (defined as the ability of the environment to provide conditions appropriate for individual and population persistence [3]) affect the dynamics of biological communities while allowing for a direct connection between the performance of organisms and their influence on ecosystem functioning [4, 5]. Habitat degradation (i.e., deterioration of habitat quality [3]) has an important impact on species traits in the biological communities, mostly affecting species with a specialized combination of traits, which are adapted to specific habitat requirements, whereas species with a more generalized combination of traits are likely to be better adapted to altered conditions [6].

This rationale has led to increasingly consider the use, value, and range of species traits to estimate their functional diversity and to link the human impacts to ecosystem functioning [7–9]. Thus, functional diversity metrics are the most employed and powerful measure to evaluate the variation of species traits within communities [10–12]. Among these metrics, functional richness (i.e., the range of ecological niches occupied by an assemblage), functional dispersion (i.e., the mean distance of individual species to the centroid of all species in the community), and functional evenness (i.e., the regularity of the distribution at which these niches are represented) are the most widely used indices [13–15].

A frequently held baseline assumption is that functional diversity decreases in response to habitat degradation and therefore it is expected that functional diversity will be higher in less disturbed habitats. However, functional diversity metrics comprise different and independent aspects within a community that respond differently to various habitat effects [16]. Although some studies have reported a higher functional diversity in better-preserved habitats, other investigations have found no differences regardless of the level of habitat quality, and some have even shown a higher functional diversity in degraded habitats [17–19]. For instance, in most cases, functional richness, functional evenness or functional dispersion are generally expected to decrease with habitat degradation, but they can also remain without significative changes when functionally redundant species may buffer species loss or, in other cases, may even be higher with habitat change when some species can fill unoccupied niches in the transformed habitats, favoring an increase in the use of resources and the occurrence of a particular set of functional traits associated with degraded habitats [19–21].

In the marine realm, coral reefs are highly diverse ecosystems encompassing different habitats and microhabitats, thus providing an excellent opportunity to test how habitat degradation affects the functional diversity of their associated communities [22]. However, coral reefs have also undergone some of the most severe ecological and structural changes at regional and global scales. For instance, human activities are increasingly degrading habitats, leading to the loss of specialist species and favoring the increasing prevalence of generalist species [6, 23]. Habitat degradation also has caused the biotic homogenization through changes in community structure, and an increase in the taxonomic and functional similarity of reef fish communities over time [e.g., 24].

Global analyses have shown consistent patterns in the functional diversity of reef-associated fish assemblages. Across ecoregions, reef fish assemblages have shown a similarity in functional trait composition [25, 26]. These large-scale patterns raise the question of whether the same stability in fish functional diversity can be also observed at smaller spatial scales. This is particularly relevant in the Caribbean because the widespread habitat coral reef degradation across the region could be favoring the relative increase of functional similarity and simplification of fish communities at smaller scales due to a species turnover among previously

differentiated fish assemblages [27]. In this sense, functional diversity indices have been shown to be sensitive enough to capture the spatial variability of fish communities, with fine-scale measures of habitat (e.g., benthic complexity, depth, hard coral cover) as important variables explaining functional patterns of local reef fish communities [22, 28, 29]. In the present study, we used an ecological trait-based ordination analysis to quantify the functional diversity (determined by the functional composition, functional richness, functional evenness, functional dispersion, and functional originality) of fish communities on two shallow Caribbean coral reefs similar in size, depth, and distance from shore, but with contrasting levels of habitat quality. There is evidence that reef habitats dominated by complex coral growth forms (such as branching forms) may favor the ecological niche (defined by Clavel et al. [30] as "all that a species requires for its population viability in a given environment, as well as its impacts on that environment") of particular reef fish species than those with corals that provide less structural complexity [22, 31, 32]. Therefore, we expected that the highly complex reef habitats constructed by key reef-building coral species in the less degraded reef would host a diverse fish trait composition and exhibit a higher functional diversity compared to the less complex reef habitats in the more degraded reef.

## Materials and methods

### Study area

The study was conducted in the Puerto Morelos reef system located on the northeastern coast of the Yucatán Peninsula, Mexico. This area has been a Marine Protected Area (MPA) since 1998 and encompasses an extended fringing reef system consisting of a series of reef units and patches that differ in size and structural complexity, separated from the coast by a shallow (<5 m) reef lagoon [33, 34]. The present study was conducted on two coral reef units located in the northern portion of the system: Bonanza reef (centered at 20˚57.6' N, 86˚48.9' W) and Limones reef (centered at 20˚59.1' N, 86˚47.9' W). These reefs are separated from one another by ~2 km and are similar in size (~1.5 km in length), back-reef depth (<5 m), and distance from the coast (<1.5 km) but with different habitat quality.

It is well-known that *Acropora palmata* is the most structurally complex in the Caribbean and previous studies have identified Limones as the reef site with the highest structural complexity and physical functionality among the northern Caribbean coral reefs [35, 36]. Bonanza has been described as a heavily degraded reef at least since the mid-2000s [37]. The live coral cover on Bonanza gradually declined from 33% in 1985 [37], to 12% by 2006–2007 [38], to less than 10% by 2015, when it exhibited a predominance of erect macroalgae (>30% cover) and extensive areas of relic *Acropora* skeletons [34, 39]. In contrast, Limones has a high coral cover (>30%), with an abundant and healthy population of the reef-building coral *A. palmata* [36] and less erect macroalgae compared to Bonanza [37, 40; Fig 1]. Both reefs are protected as part of the Puerto Morelos Reef National Park since 1998 and were open to visitation until 2014, when Limones was decreed a critical habitat for the conservation of *A. palmata*. Since then, Limones has been closed to visitation, but Bonanza was already in a degraded state, it has remained open for visitation [36].

Previous studies conducted in Bonanza and Limones have investigated the potential effects of habitat degradation on several reef-associated communities. Despite notable differences in the level of habitat degradation between both reefs, their fish and invertebrate communities have shown a great similarity in patterns of dominance and trophic webs [34, 39, 40]. However, the isotopic niche and the trophic level of a specialist species (the spotted spiny lobster, *Panulirus guttatus*) differed between both reefs [39].

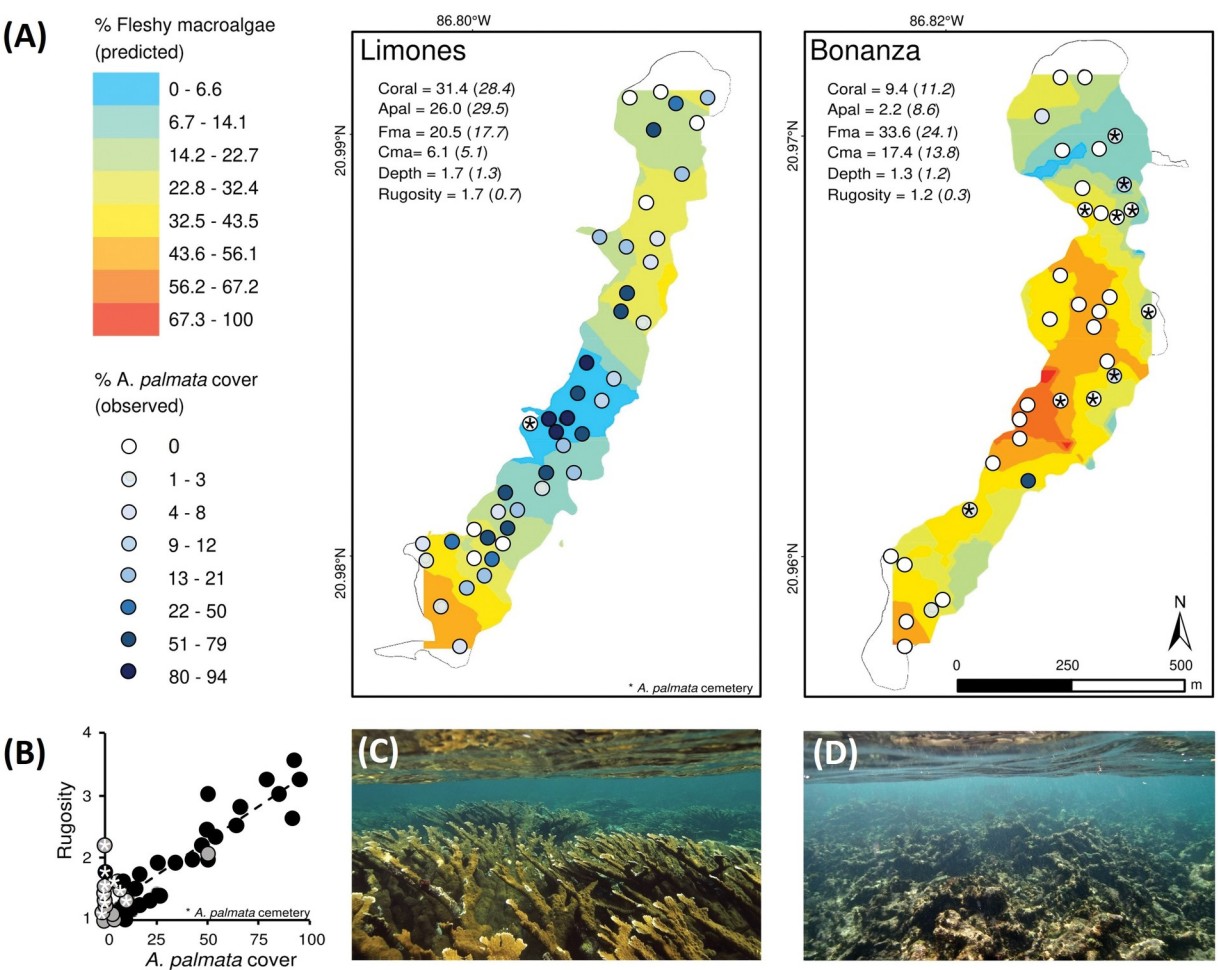

**Fig 1. Contrasting benthic habitat condition in adjacent shallow coral reefs.** (A) Kriging interpolation of the cover of fleshy macroalgae and the percent cover of *Acropora palmata* in randomly selected transects in Limones and Bonanza reefs. The Kriging interpolation was used, as the cover of fleshy macroalgae was spatially autocorrelated among transects. For each reef the mean (standard deviation) cover of corals (Coral), cover of *A. palmata* (Apal), cover of fleshy macroalgae (Fma), cover of calcareous macroalgae (Cma), depth and rugosity index were calculated. All reported habitat variables were significantly different between reefs (Welch's t-test p < 0.01 in all cases). Depth was statistically similar (Welch's t-test p = 0.21); (B) The relationship between transect-level cover of *A. palmata* and rugosity in Limones (black dots; n = 44) and Bonanza (grey dots; n = 35) reefs. The overall relationship (dotted line) among these variables is strong and significant (R² = 0.84; p < 0.001). (C) Highly complex *A. palmata* reef representative of the central Limones reef and (D) Degraded coral reef habitat in Bonanza reef. * in (A) and (B) indicates transects on areas of *A. palmata* standing skeletons. Source for photos: L. Alvarez-Filip.

## Reef benthic composition

Benthic habitat composition and structural complexity was described using 79 randomly selected 10 m-long transects throughout the back reef and crest zones on both reefs (i.e., 35 transects on Bonanza and 44 throughout Limones, Fig 1). The spatial extent of the shallow reef-habitats (< 5m) and standing patches of live *Acropora palmata* of Limones and Bonanza reefs were characterized by means of manta tows. One or two observers were towed behind a small boat holding a GPS (inside a thin plastic cage). Resulting GPS-tracks were used to define the limits of each reef (>5 m depth). Spatial representation and calculation were carried out in ArcGIS 10.3.

Fifty points were randomly generated using the spatial extent of shallow habitats of each reef. Each point has an associated latitude and longitude that represent the start of the transect;

the direction (N or S) was randomly chosen in situ by flipping a coin. At each transect, the cover of the main components (live coral species, turf algae, fleshy macroalgae, calcareous macroalgae, among others) was measured by means of the point-intercept method, following the protocol of the Atlantic and Gulf Rapid Reef Assessment (AGRRA) for benthic components [41]. A surveyor recorded the benthic component intercepting the line every 10 cm (i.e., 100 points per transect). The cover of each benthic component was estimated as a percentage of the number of points overlaying the transect. In addition, along each transect, reef structural complexity was measured using the rugosity index, which is the ratio of a length of a chain molded to the reef surface to the linear distance between its start and endpoint [42]. A perfectly flat surface would have a rugosity index of one, with larger numbers indicating more complex surfaces. To achieve this, a 35-metre chain (0.5 cm link-length) was used along the 10 m length of the transect.

## Reef fish composition

To survey the reef fish communities, underwater visual censuses were haphazardly performed on Bonanza and Limones. We defined the Bonanza and Limones reef communities as the result of multiple censuses sampled randomly at each site, consistent with the scale at which fish communities can respond to fine-scale measured benthic habitat components (S3 Table). Thus, a total of 101 belt-transects of 60 m$^2$ (30 m × 2 m) were obtained on both reefs (47 transects in Bonanza and 54 in Limones). All transects were surveyed between 9:00 and 15:00 h. At each transect, all conspicuous fishes were identified and counted, and their total length was estimated in cm. Individual fish size was assigned to 10 cm categories (from 10 to 150 cm). In addition, fish biomass was estimated using the length-weight relation: $Weight = a*Length^b$, where the coefficients $a$ and $b$ were obtained from FishBase [43]. We acknowledge that underwater visual censuses are not the best method to register larger home ranging or territorial species and may underestimate the number of species and density of cryptobenthic species. Moreover, the difficulty of accurately assessing and sampling cryptobenthic species has resulted in a poor body of quantitative data of this assemblages [44], both regionally and globally, making difficult to include them in trait-based approaches. To reduce the observer bias to the minimum, all visual censuses were conducted by trained scientific observers that have jointly carried out similar surveys in the Caribbean for several years. Count time was not standardized because it was dependent on fish abundance and species diversity.

## Biological traits of reef fishes and functional diversity metrics

We selected six categorical traits that describe multiple aspects of fish ecology and have been used in previous studies to examine biogeographic patterns and impacts on ecosystem functioning at regional and global scale [26, 45]. The traits were: 1) fish body size, 2) home range, 3) period of activity, 4) gregariousness, 5) position in water column, and 6) diet (Table 1). These traits were compiled from Quimbayo et al. [46] and categorized based on the criteria used by Mouillot et al. [25]. Although the use of discrete trait values represents a coarse categorical approximation of the ecological function of organisms, these traits capture a broad range of "responses" and "effects" to environmental gradients and have been shown to be sensitive to habitat disturbances [47, 48]. For example, the availability of shelter following habitat degradation may affect the community structure of reef fishes in terms of their body size or promote the migration of fishes to relatively unaffected habitats in relation to their home range and diet requirements through trophic interactions with other food-web components (S1 Table).

**Table 1. Functional traits used to calculate fish functional diversity on coral reefs (according to Quimbayo et al. 2021).**

| Type | Trait | Description and categorization |
|---|---|---|
| Morphological Categorical (ordinal) | Fish body size | Maximum total length of each fish species was coded using six categories: 0–7 cm; 7.1–15 cm; 15.1–30 cm; 30.1–50 cm; 50.1–80 cm; and >80 cm. |
| Behavioral Categorical (ordinal) | Home range | Spatial extent at which fishes are moving was coded using three categories: sedentary-territorial species (species that move less than a few 10 m²); mobile within-reef (species that can move 10s of meters within the same reef); and very mobile among reefs (species that can move distances over 100 m² and usually move between habitats/reefs). |
| Behavioral Categorical (ordinal) | Period of activity | Period of the day at which fishes are active was coded using three categories: diurnal species; diurnal-nocturnal species (i.e., crepuscular); and nocturnal species. |
| Behavioral Categorical (ordinal) | Gregariousness | The social strategy adopted by fishes to minimize predation and energetic cost while feeding was coded using four categories: solitary; pairs; small groups (3 to 20 individuals on average); medium groups (21 to 50 individuals on average); and large groups (>50 individuals on average). |
| Behavioral Categorical (ordinal) | Position in water column | The level of water column occupied by fishes was coded using three categories: benthic; benthopelagic; and pelagic. |
| Ecological Categorical (nominal) | Diet | The main items consumed by each fish species was coded using seven categories: herbivore-detritivore; macroalgal feeder; invertivores of sessile invertebrates (e.g., corals, sponges, and ascidians); invertivores of mobile benthic invertebrates (e.g., crustaceans, annelids, echinoderms, and mollusks); planktivores; omnivores; and piscivores. |

## Data analysis

To characterize the reef habitat condition of Bonanza and Limones, we used the information of coral cover, cover of *A. palmata*, reef rugosity, cover of macroalgae and cover of crustose coralline algae at the transect level. Using the kriging interpolation in ArcGIS 10.3, we explored the likelihood of creating spatially interpolated maps of the various benthic components. Kriging is a geostatistical spatial interpolation method that models the relationship between the distance and variance of sampled points to predict values at unsampled locations. The Kriging method was chosen as it was the most likely to give accurate outputs given the numerous transects from which we were interpolating (see Meng et al. [49]). Spatial correlation between transects was assessed with semi-variograms.

Fish assemblages were characterized ecologically using three univariate traditional metrics that describe the overall fish community structure: the species richness (number of species/60 m²), the density of individuals (number of individuals/60 m²), and the biomass (g/100 m²). The three-community metrics (mean ± standard deviation) were compared between reefs at the transect level using Welch's *t*-tests with a significance level of 95%. Moreover, to assess the functional variation of reef fish assemblages, we first calculated the pairwise distances between species traits using Gower´s distance metric and then we performed a principal coordinate analysis (PCoA) for the construction of a synthetic four-dimension functional space within which the species were arranged based on their trait values (S2 Fig).

We calculate the ecological and functional metrics at the transect level to have a significant replication and a finer scale when comparing the response of fish communities with their habitat quality on both reefs. Thus, variation in the functional composition of fish traits was assessed using average values of the proportions of each fish trait category by computing the community-level weighted means of trait values (CWM) using function "functcomp" from the R package "FD" [50, 51]. In addition, we used the functions "dbFD", and "multidimFD" from the R package "FD" to calculate four complementary indices that measure different aspects of functional diversity on fish assemblages: functional richness (FRic); functional dispersion (FDis); functional evenness (FEve); and functional originality (FOri) [13, 14]. Finally, to assess the functional change between reefs we use the function "FSECchange" (see details on Mouillot et al. [15]) to calculate the turnover in the functional structure of fish communities, estimating the dissimilarity of functional measures between Bonanza and Limones.

We chose these measures of functional diversity because they consider independent facets of functional diversity within the community, can incorporate multiple categorical and numerical traits in multidimensional trait space [13, 14, 52], and are the most widely used indices in studies that consider functional ecology to evaluate changes on fish communities. Functional richness is positively correlated with the number of species present and is independent of species abundances; however, it is the only index that reflects the range of trait values, that is, how much of the functional niche space is filled by the occurring species [13]. Functional dispersion and functional evenness are unaffected by species richness, are not strongly influenced by outliers, and account for species abundances [14], whereas functional originality may represent an important indicator of an assemblage's functional redundancy [7].

## Results

### Reef benthic composition

Bonanza and Limones exhibited considerably distinct benthic composition and structural complexity. Reef rugosity, total coral cover (reef-building species + *Millepora* spp.), and the cover of *A. palmata* were significantly higher in Limones (Fig 1A), whereas fleshy and calcareous macroalgae cover were the dominant benthic components in Bonanza (Fig 1A). In particular, we found around three times more coral cover in Limones, of which more than 80% was *A. palmata*. In contrast, this species only represented around 20% of the total coral cover in Bonanza (the more degraded reef). Fleshy macroalgae cover was the only benthic component that showed spatial autocorrelation within transects, allowing to conduct an interpolation to spatially represent the cover of this group across each reef (Fig 1A). Overall, the relationship between the cover of dominant coral *A. palmata* and reef rugosity was strong and positive, indicating that the high abundance of this species largely explains the high habitat structural complexity observed in Limones (Fig 1B–1D).

### Reef fish composition

In total, 68 fish species belonging to 21 families were recorded on Bonanza and Limones reefs (S2 Table). Bonanza presented lower values of species richness, density of individuals, and biomass (Fig 2). The largest difference in these descriptors corresponded to species richness, with the number of species by unit of area being nearly 30% higher in Limones than in Bonanza (mean ± standard deviation: $14.87 \pm 3.37$ vs $9.90 \pm 3.13$; $t = 7.53$, df = 94.35, $p < 0.001$). Fish density (Limones: $6.34 \pm 0.62$, Bonanza: $5.08 \pm 0.73$; $t$-test on log-transformed data, $t = 9.04$, df = 84.60, $p < 0.001$) and fish biomass (Limones = $12.82 \pm 1.17$; Bonanza = $11.72 \pm 1.35$; $t$-test on log-transformed data, $t = 4.23$, df = 85.82, $p < 0.001$) also showed significant differences between reefs (Fig 2).

### Species traits and multidimensional functional space occupied by reef fishes

The multidimensional space constructed with the trait distribution of the 68 fish species in Bonanza and Limones shows that the four first dimensions of the PCoA cumulatively explained 78.9% of the distribution of fish species traits. Axes 1 and 2 of the PCoA accounted for 50.7% of the total variation, while axes 3 and 4 accounted for 28.2%. Trait categories of fish body size, home range, gregariousness, and water column position better separated the trait categories along axes 1 and 2, whereas the period of activity and diet did so along axes 3 and 4 of the PCoA (S1 Fig). Fish body size and home range increased from left to right along the first axis (S1A and S1B Fig), while gregariousness and water column position changed along the

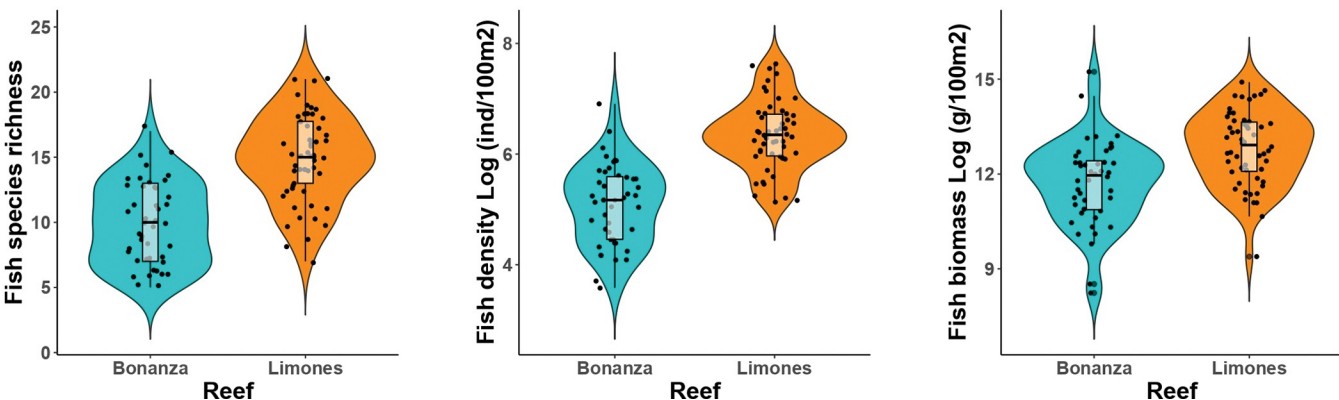

**Fig 2. Variation in ecological indicators of reef fish assemblages between the coral reefs Bonanza and Limones.** Violin plots show: (A) the species richness (number of species/60m$^2$), (B) density (individuals/60 m$^2$), and (C) biomass (g/100 m$^2$). Boxplots display medians (black horizontal line) upper and lower quartiles (boxes), and minimum and maximum ± 1.5 interquartile range (IQR) (whiskers). All ecological fish descriptors were significantly different between reefs (Welch's *t*-test, p < 0.001 in all cases). Density and biomass values were log-transformed to fit a normal distribution.

second axis (S1C and S1D Fig). Period of activity changed along the third axis of the PCoA, but the greatest proportion of the functional space was filled by diurnal species towards the left side (S1E Fig). Diet showed a mixed distribution along axes 3 and 4, where herbivore-detritivores were positioned in the middle-top of the functional space, while invertivores (feeders of sessile and mobile benthic invertebrates) and omnivores were found mainly in the central and bottom area. Planktivores were positioned at the bottom-right while piscivores were positioned at the middle-left of the functional space (S1F Fig).

The functional composition of fish assemblages, calculated by community weighted trait means (CWM), exhibited the proportion of each trait category between Bonanza and Limones (Fig 3). We found an apparently consistent fish trait composition between reefs despite their contrasting levels of habitat degradation and significant differences in number of fish species, density, and biomass, i.e., the same fish trait categories were dominant in Bonanza and Limones. On both reefs, medium-sized species (15.1–30 cm) represented ~40% of the total abundance in fish body size trait (Fig 3A), mobile species accounted for almost 80% of the total abundance in home range trait (Fig 3B), diurnally-active species represented 83% of the total abundance in period of activity (Fig 3C), species that form medium groups up to 50 individuals accounted for ~40% of total abundance in gregariousness trait (Fig 3D), over 95% of the total abundance in water column position was accounted for by fishes with benthopelagic habits (Fig 3E), and herbivores-detritivores, together with mobile invertebrates feeders, represented ~70% of the total abundance in diet categories (Fig 3F).

## Functional diversity metrics

The functional diversity indices employed to compare the reef fish communities exhibited different responses between reefs (Fig 4). Functional richness was significantly lower in Bonanza compared to Limones (0.18 ± 0.11 vs 0.31 ± 0.11; *t*-test on square root transformed data, $t = 6.12$, df = 91.32, p < 0.001) whereas, conversely, Bonanza exhibited higher values of functional evenness than Limones (0.63 ± 0.11 vs 0.55 ± 0.08) resulting in significantly differences between reefs ($t = 3.69$, df = 82.58, p < 0.001). In contrast, no differences were observed in functional dispersion (0.50 ± 0.11 vs 0.49 ± 0.07; $t = 0.84$, df = 72.33, p = 0.39) or functional originality between Bonanza and Limones (0.15 ± 0.09 vs 0.17 ± 0.07; $t = 1.18$, df = 83.54, p = 0.23).

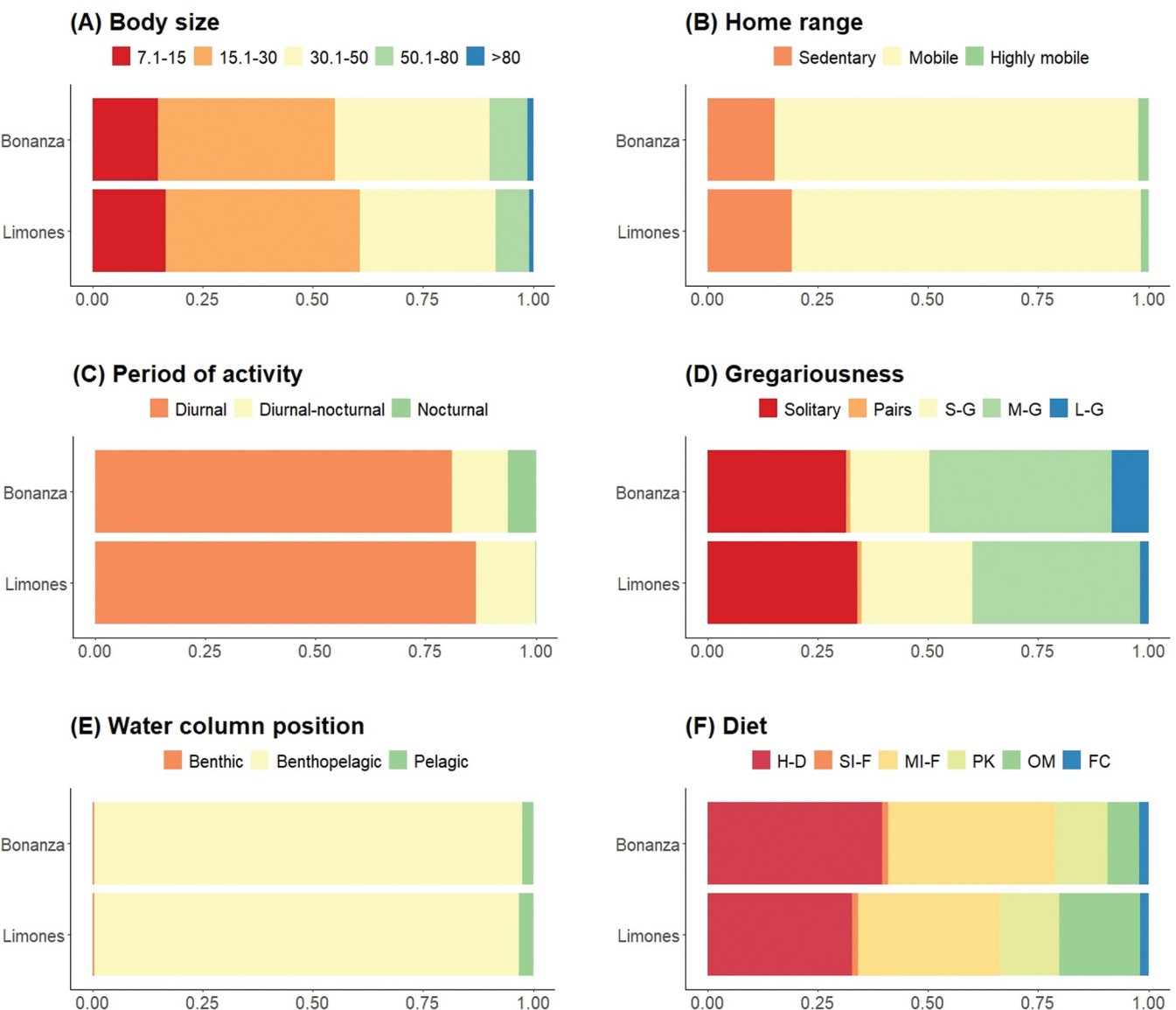

**Fig 3. Comparison of the fish trait composition in Bonanza and Limones coral reefs.** Stacked bar charts show the fish relative abundance of each trait category calculated from the community-level weighted means of trait values (CWM) for a set of 68 fish species. In (D) gregariousness: S-G (small groups), M-G (medium groups), L-G (large groups); (F) Diet: H-D (herbivores-detritivores), SI-F (sessile invertebrate feeders), MI-F (mobile benthic invertebrate feeders), PK (planktivores), OM (omnivores), and FC (piscivores).

Changes in the functional richness (i.e., the convex hulls represent the portion of the functional space filled for each fish communities) showed substantial overlap between Bonanza and Limones (Fig 4A). According to axis 1 and 2 of the PCoA, the functional space filled by functional traits of Bonanza represented 83% of the total volume while the functional space represented by functional traits of Limones represented 85%. In axis 3 and 4, we found a relatively greater differentiation, where the functional space filled by Bonanza was of 74% while in Limones it represented 89% of the total functional trait distribution. The overall shift in the functional trait space among reefs estimated by the portion of the functional space filled only by Bonanza and filled only by Limones represented 26% of their combined volume in axis 1 and 2 and 31% in axis 3 and 4 (Fig 4A).

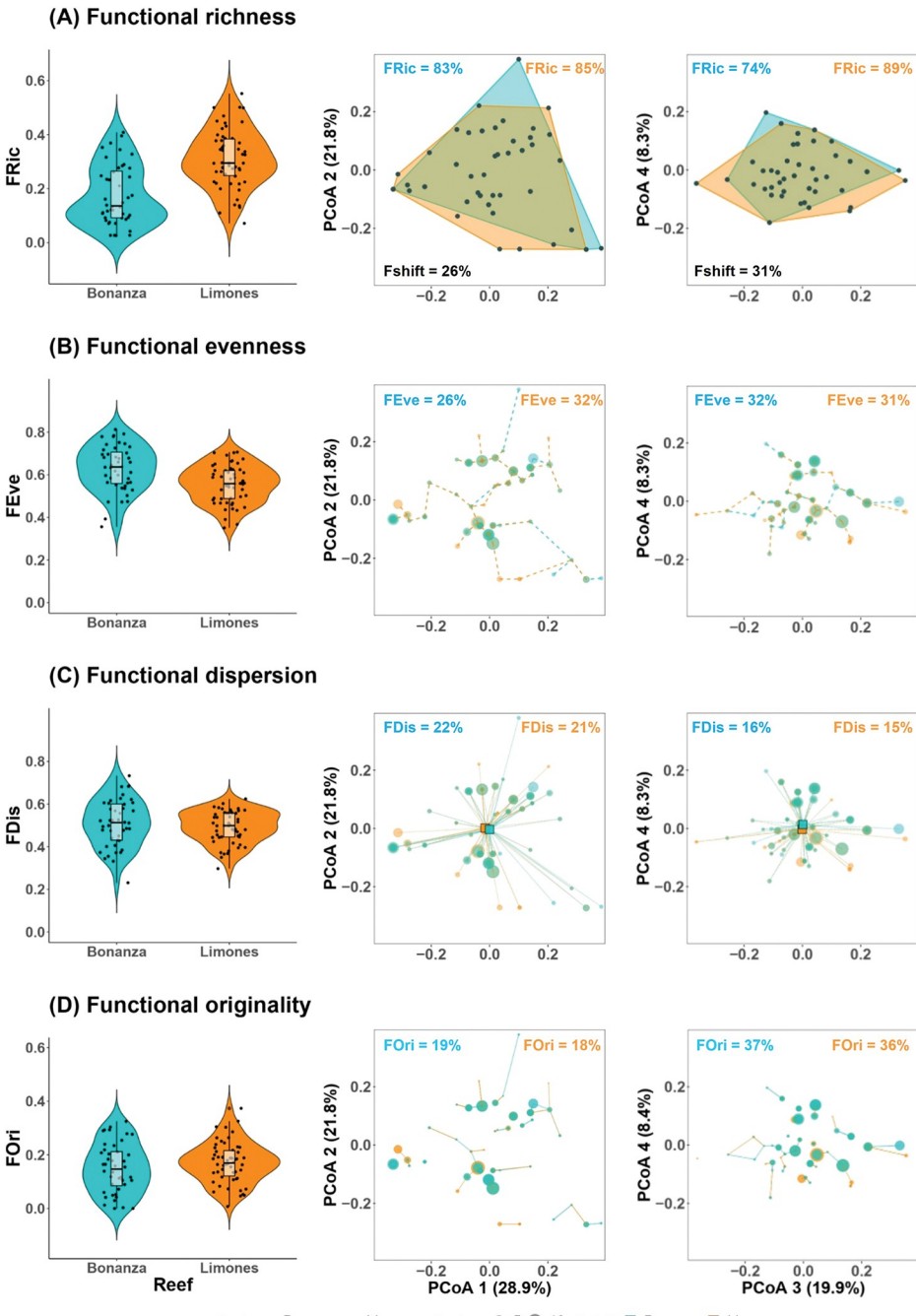

**Fig 4. Variation in functional diversity descriptors of reef fish communities in Bonanza and Limones.** The left panel shows Violin plots of (A) functional richness (FRic), (B) functional evenness (FEve), (C) functional dispersion (FDis), and (D) functional originality (FOri) in Bonanza (filled in turquoise) and Limones (filled in orange). Boxplots inside displays medians (black horizontal line), upper and lower quartiles (boxes), and minimum and maximum ± 1.5 interquartile range (IQR) (whiskers). FRic and FEve were significantly different between reefs (Welch's t-tests, $p < 0.001$) whereas FDis and FOri were not (Welch's t-tests, $p > 0.05$). FRic values were square root transformed to fit a normal distribution. The central and right panel show the trait distribution in Bonanza (filled in turquoise) and Limones (filled in orange) fish communities for functional richness, functional evenness, functional dispersion, and functional originality in the first four axes of a Principal Coordinate Analysis. The centroids in functional dispersion are indicated with squares for Bonanza (filled in turquoise) and Limones (filled in orange).

Changes in functional evenness (i.e., the modifications in the regularity of abundance distributions in the functional space along the shortest minimum spanning tree linking all the species) were visually less evident, with no important shifts at the community level despite significant differences between reefs (Fig 4B). Likewise, functional dispersion (abundance-weighted mean distance relative to the centroid of the fish community) and functional originality (average pairwise distance between a species and its nearest neighbor) were very similar, showing that the percentage of the maximal pairwise distance observed between reefs did not change along all axes of the PCoA (Fig 4C and 4D).

## Discussion

Our findings suggest that habitat condition does not appear to influence trait composition and functional redundancy of Caribbean reef fishes at small-spatial scale. Despite notable differences in hard coral cover, algae cover, and habitat structural complexity as well in the number of species, density, and biomass of fishes between Bonanza and Limones, there was a striking similarity in the composition and diversity of fish functional traits between reefs. Differences in functional richness and evenness are likely to result from the higher number of species and individuals associated with a greater habitat complexity provided by the stands of *A. palmata* in Limones. However, despite its higher structural complexity, Limones did not host species with different functional traits. This is likely because most Caribbean fish species are not strict habitat specialists [27, 53] and both studied reefs are subject to similar environmental variability and pressures.

We can hypothesize at least two possible causes that are not mutually exclusive to explain the similarity in fish trait structure observed at local scales between reefs with contrasting levels of habitat degradation. First, global analyses have shown remarkably consistent patterns of functional fish trait composition in temperate and tropical marine realms worldwide, with fish communities exhibiting a common structure of traits despite differences in species richness, environmental regimes, and habitat types [25, 26]. Therefore, the similarity in fish trait composition and the maintenance of a common structure observed in our study possibly reflect those larger-spatial scales patterns. This suggests that environmental and ecological pressures (rather than habitat complexity) maintain most of the key trait combinations. This could be acting as an environmental filtering in which local trait composition is a result of those traits present at a larger spatial scale [25, 26, 54]. Nevertheless, although a similar proportion of fish trait categories was displayed between reefs, Bonanza (the more degraded reef) exhibited a significantly lower density and biomass of fish individuals compared to Limones (S3 Fig), which implies that Bonanza may be more vulnerable to a decline in fish trait diversity if degradation persists in the future or if local pressures such as overfishing increase [55].

Second, the spatial regional context in terms of availability of species and the different states of habitat condition might also largely contribute to explain the observed spatial patterns (Fig 5). Both Bonanza and Limones are located under the same spatial regional context of a larger-scale widespread degradation of reef structure, which has been documented in the Caribbean [35, 56]. This context is probably exerting a stronger influence, limiting the variety, range, and type of functional traits that can be present at local scales, even on fish communities that inhabit reefs with a good habitat condition such as Limones. To further explore this, we compared the functional trait diversity of the fish assemblages observed only on Limones and Bonanza reefs with the functional trait diversity found in the entire region. For this, we used data from 20 additional sites surveyed between 2016 and 2019 over a depth range from 1 to 12 m, encompassing a ~40 km-long linear extent of reefs along the northern portion of the Mexican Caribbean (S4 Fig). We found that despite the considerably larger number of species

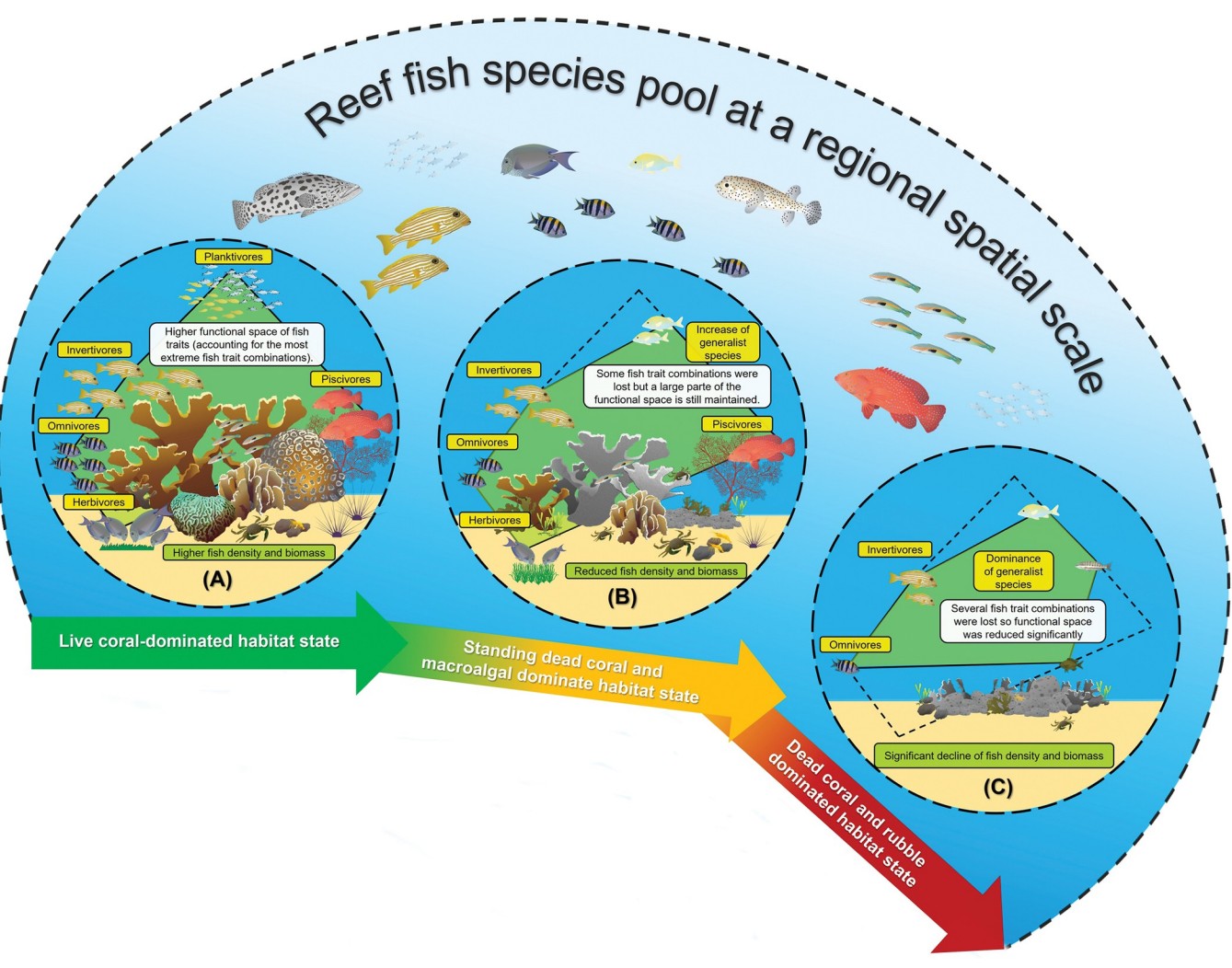

**Fig 5. Conceptual diagram displaying the alternate states in functional trait composition on shallow coral reefs at different levels of habitat degradation.** From left to right: (A) live coral-dominated habitat state with higher structural complexity, (B) standing dead coral and macroalgal- dominated habitat state with lower structural complexity, and (C) dead coral and rubble-dominated habitat state with virtually no structural complexity. The green polygons represent the functional space filled by the fish species with the most extreme trait combinations on each reef. Despite the notable differences in the benthic habitat states, number of fish species, and density of individuals between (A) and (B), the functional space represented in (B) is still relatively similar in maintaining key ecological functions as that in (A), probably due to an increase in the availability of resources (increased algal cover and density of macro-invertebrates because of a greater refuge density in dead corals) that benefit some fish species, resulting in ecological trade-offs against habitat degradation, although fish communities become more vulnerable to species loss due to their lower density. Finally, if habitat degradation continues in the future such as shown in (C), the subsequent loss of branching corals and the total loss of structural complexity, will likely result in a drastic decrease in functional space (i.e., a reduced trait diversity) due to the loss of entire key ecological functions in the system. Source of vectors: Integration and Application Network (ian.umces.edu/media-library).

recorded at a regional-scale (2.2-fold the species richness observed at Bonanza and 1.8-fold the species richness observed in Limones), the regional functional space based on shared trait diversity was very similar to the ones obtained only for Limones and Bonanza (S5 Fig). On average, ~75% and ~85% of the regional functional space was filled by the fish species represented by the functional space of Limones and Bonanza, respectively (S5 Fig). The current study focused mainly on conspicuous reef fishes in shallow coral reefs, where large and highly

mobile fish species are generally not abundant as they move to deeper areas. Also, underwater visual censuses underestimate the number and density of cryptobenthic species, resulting in a scarcity of quantitative data on this assemblage, both regionally and globally [57], making it difficult to include them in trait-based approaches. Furthermore, our regional assessment included species that are not found in shallow waters or that inhabit outer reefs (e.g., *Chilomycerus reticulatus* or *Chromis insolata*); therefore, it is likely that the overlap in the functional space of our fish assemblages on Bonanza and Limones relative to the spatial regional context would probably be even larger.

The widespread coral reef degradation in the Caribbean may have important consequences for fish productivity and ecosystem functioning following the coral mortality [58]. Nevertheless, there is evidence that degraded reefs with diminished levels of structural complexity can still support some degree of productivity, being fairly robust to the initial stages of reef degradation [58, 59]. In degraded reefs, refuge density declines but prey vulnerability increases, leading to an initial rise in resource availability and productivity for a significant part of the reef fish community, particularly herbivores and invertivores [58, 60]. Furthermore, habitat degradation permits the exploitation of novel resources by fishes that feed on macroinvertebrates [7]. Indeed, macroinvertebrates such as crustaceans are important contributors to the productivity of coral reefs, and dead coral structures can still be an important microhabitat for these invertebrate communities, supporting up to three orders of magnitude more individuals than living corals [61]. For the same reefs studied here, the diversity and abundance of macro-crustaceans were higher on Bonanza (i.e., the more degraded reef) compared to Limones, with the coral rubble and stands of *Acropora* skeletons constituting the microhabitats most occupied by these organisms [40]. In addition, Briones-Fourzán et al. [39] found that, on Bonanza, spiny lobsters showed a preference for feeding on crustaceans and mollusks, which in turn feed on fleshy, calcareous, and coralline macroalgae, benthic constituents that have become increasingly prevalent on degraded reefs such as Bonanza [34]. These results suggest that habitat conditions on coral reefs such as Bonanza, characterized by the dominance of algae cover and dead coral skeletons, may offset the initial stages of habitat degradation by increasing the resource availability for some groups of herbivores and invertivores (such as mobile invertebrates and fishes). However, this may only occur in the short term because, if degradation persists, all structural complexity could be lost, followed by a decline in fish productivity and their functioning.

In contrast, healthier, coral-dominated reefs, especially with branching or large, massive life-forms, have been positively correlated with a higher habitat structural complexity that supports greater richness, abundance, and biomass of associated reef fish assemblages than degraded, algal-dominated coral reefs [31, 32, 62]. Previous studies on Caribbean coral reefs have shown that high structural complexity is an important predictor of fish functional diversity because it supports a rich variety of physical habitats and niches that increase the functional richness of fishes on these reefs [29, 63].

Our findings suggest that the significant differences in fish functional richness and functional evenness detected at finer spatial scales (i.e., at the transect level) could also be explained by the greater habitat complexity provided by *A. palmata* on Limones and the higher number of fish species, which may support a greater diversity and range of niches or functional strategies than a degraded coral reef habitat with less structural complexity such as Bonanza. Likewise, functional evenness may be most relevant at finer spatial scales at which the biomass or density of individuals of different species could buffer against progressive habitat degradation [28]. Thus, functional richness, which is strongly influenced by the number of species, reflected a higher trait diversity in Limones, whereas functional evenness, which is rather influenced by the density of individuals, showed a more regular distribution of trait combinations in Bonanza [13, 14]. Yet, when

comparing the functional richness at the reef spatial scale, we found over 80% overlap between the functional space of both reefs, reflecting a similarity in functional niche. These results suggest that most fish trait combinations found on Limones are still maintained on Bonanza despite the contrasting levels of habitat degradation between these reefs.

Therefore, the significantly higher structural complexity provided by the dominance of *A. palmata* in Limones and the associated greater diversity and density of fishes did not necessarily contribute to a better complementarity of species traits compared to Bonanza. Thus, it is likely that fish species with higher relative abundances on Bonanza share trait combinations with fish species with higher relative abundances on Limones and hence pairwise distances among fish species in the functional trait space were very similar between reefs. In addition, the loss of functional originality and functional dispersion following habitat degradation can also occur if species with a particular trait combination are selectively removed from the assemblage, resulting in smaller pairwise distances among species traits [7].

Our results are consistent with previous studies showing that widespread habitat degradation in many Caribbean coral reefs has increased the functional similarity of fish communities (i.e., the biotic homogenization) as a result of the loss of habitat structural complexity and changes in the composition and diversity of coral assemblages [22, 27, 35]. This is important because it suggests that changes at larger spatial scales, such as worldwide declines in specialist species and their replacement by generalist species, may increase the functional similarity through biotic homogenization of biodiversity at smaller spatial scales [30]. Although the dynamic responses of reef fishes to habitat degradation imply the loss of specialists at local scales, whereas generalists could be broadly favored under intensifying anthropogenic pressures [e.g., 5], the patterns observed in the present study suggest that, even in coral reefs with a good habitat condition such as Limones, the functional homogenization of fish communities can increase.

Our findings may be reflecting a functional homogenization of fish communities due to widespread degradation at a regional spatial scale [e.g., 34, 54] and a low-level of habitat specialization by the communities of Caribbean reef fishes. Moreover, regardless of the functional similarity exhibited by fish communities, the low values of functional originality observed here (see Fig 4) may be an important indicator of the functional redundancy of fish communities on Bonanza and Limones. Functional redundancy may act as biological insurance against habitat disturbances, providing at least an elementary representation of the fish functional roles on these coral reefs [7]. Although degraded reefs with low fish density such as Bonanza can support functionally diverse communities and maintain certain processes, they are far more vulnerable to species loss than healthier reef habitats such as Limones.

Overall, our results demonstrate that coral reefs with contrasting levels of habitat degradation can host functionally similar fish communities, but only in the short term because, if degradation persists, the mechanisms that maintain these functional roles could be lost, followed by a decline in the ecosystem productivity and functioning. Additional future work should consider other components underlying the loss of functional diversity in coral reef ecosystems, such as effects of habitat degradation on the communities of macro-invertebrates or fishing pressure at larger spatial scales. Such studies will be critical for determining the local-scale dynamics that affect the functional structure of associated fish communities and for understanding the role of trait diversity in maintaining the suite of ecosystem functions and services in the face of widespread degradation of reef structure.

## Supporting information

**S1 Fig. Multidimensional functional space shows the distribution of convex hulls for each fish trait categories (represented by different colors) in Bonanza and Limones coral reefs.**

Figures A-D display the convex hulls for body size, home range, gregariousness, and water column position traits in PCoA 1 and PCoA 2, while figures e-f display the convex hulls for activity and diet along PCoA 3 and PCoA 4. In (C) gregariousness: S-G (small groups), M-G (medium groups), L-G (large groups); (F) Diet: H-D (herbivores-detritivores), SI-F (sessile invertebrate feeders), MI-F (mobile benthic invertebrate feeders), PK (planktivores), OM (omnivores), and FC (piscivores).
(TIF)

**S2 Fig. Quality of the functional space.** The comparison with an UPGMA dendrogram is also shown. The mean squared deviation (mSD) is used to assess the quality of the functional space.
(TIF)

**S3 Fig. Comparison of the fish trait composition in Bonanza and Limones coral reefs.** Bar charts show the fish absolute abundance of each trait category calculated from the community-level weighted means of trait values (CWM) for a set of 68 fish species. In (d) gregariousness: S-G (small groups), M-G (medium groups), L-G (large groups); (f) Diet: H-D (herbivores-detritivores), SI-F (sessile invertebrate feeders), MI-F (mobile benthic invertebrate feeders), PK (planktivores), OM (omnivores), and FC (piscivores).
(TIF)

**S4 Fig. Location of the 20 surveyed coral reef sites.** The study sites are in the northern portion of the Mexican Caribbean (including the Puerto Morelos reef system).
(TIF)

**S5 Fig.** Multidimensional functional space filled by fish assemblages in Bonanza (upper panel) and Limones (bottom panel). The black polygons represent the functional space occupied by all fish species present on 20 reef sites of the Puerto Morelos reef system. The turquoise-filled polygon represents the functional space occupied by fish trait combinations in Bonanza while the orange-filled polygon represents the functional space occupied by the fish trait combinations in Limones. Black points depict the reef fish species outside of the functional space covered by the polygons of Bonanza and Limones.
(TIF)

**S1 Table. Functional traits selected for this study to explore changes in functional structure of Caribbean reef fishes.** We selected six traits that describe several facets of fish ecology and are available for many species of tropical regions. These traits were employed to perform the functional diversity indices and they have been shown to have ecological implications for ecosystem functioning and a range of responses and effects by the coral reef fishes against habitat changes.
(PDF)

**S2 Table. List of 68 species and 21 families of reef fishes recorded in Bonanza and Limones and their functional traits values.** Species traits were compiled by Quimbayo et al. (2021) and were coded into several categories according to Mouillot et al. (2014). Body size: 0–7 cm (s1), 7.1–15 cm (s2), 15.1–30 cm (s3), 30.1–50 cm (s4), 50.1–80 cm (s5), and >80 cm (s6). Home range: sedentary-territorial species (sed), mobile species (mob), and very mobile species (vmob). Period of activity: diurnal species (day), diurnal-nocturnal species (both), and nocturnal species (night). Gregariousness: solitary (sol), pairs (pair), small groups (smallg), medium groups (medg), and large groups (largeg). Position in water column: benthic (bottom), bentho-pelagic (low), pelagic (high). Diet: herbivore-detritivore (hd), macroalgal feeder (hm), sessile invertebrates (is), mobile benthic invertebrates (im), planktonic (pk), omnivore (om), and

piscivore (fc).
(PDF)

**S3 Table. Fish, benthos, and trait data of Bonanza and Limones reef sites.**
(XLSX)

## Acknowledgments

We thank Alba Gonzalez-Posada, Tomás López-Londoño, Kelly Gómez-Campo, Fernando Negrete-Soto and Cecilia Barradas-Ortiz for their help in conducting the fieldwork.

## Author Contributions

**Conceptualization:** Manuel Olán-González, Lorenzo Alvarez-Filip.

**Data curation:** Manuel Olán-González.

**Formal analysis:** Manuel Olán-González.

**Funding acquisition:** Enrique Lozano-Álvarez.

**Supervision:** Patricia Briones-Fourzán, Gilberto Acosta-González, Lorenzo Alvarez-Filip.

**Validation:** Patricia Briones-Fourzán, Gilberto Acosta-González, Lorenzo Alvarez-Filip.

**Visualization:** Manuel Olán-González, Lorenzo Alvarez-Filip.

**Writing – original draft:** Manuel Olán-González, Lorenzo Alvarez-Filip.

**Writing – review & editing:** Manuel Olán-González, Patricia Briones-Fourzán, Enrique Lozano-Álvarez, Gilberto Acosta-González, Lorenzo Alvarez-Filip.

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
