## [Decision Letter · Decision Letter 0]

19 Jun 2023

PONE-D-23-03979Similar functional composition of fish assemblages despite contrasting levels of habitat degradation on shallow Caribbean coral reefsPLOS ONE

Dear Dr. Alvarez-Filip,

Thank you for submitting your manuscript to PLOS ONE. After careful consideration, we feel that it has merit but does not fully meet PLOS ONE’s publication criteria as it currently stands. Therefore, we invite you to submit a revised version of the manuscript that addresses the points raised during the review process.

We look forward to receiving your revised manuscript.

Kind regards,

Frank O. Masese, Ph.D

Academic Editor

PLOS ONE

“ProgramUNAM-DGAPA-PAPIIT, project IN-205614), granted to L.A.-E.”

“We thank Alba Gonzalez-Posada, Tomás López-Londoño, Kelly Gómez-Campo, Fernando 511 Negrete-Soto and Cecilia Barradas-Ortiz for their help in conducting the fieldwork. The present 512 study was supported by Consejo Nacional de Ciencia y Tecnología (CONACYT) doctoral 513 fellowship (CVU 707432) granted to M.O.-G. This study was funded by Universidad Nacional 514 Autónoma de México (ProgramUNAM-DGAPA-PAPIIT, project IN-205614), granted to L.A.-E”

“ProgramUNAM-DGAPA-PAPIIT, project IN-205614), granted to L.A.-E.”

7. We note that [Figure 1] in your submission contain [map/satellite] images which may be copyrighted. All PLOS content is published under the Creative Commons Attribution License (CC BY 4.0), which means that the manuscript, images, and Supporting Information files will be freely available online, and any third party is permitted to access, download, copy, distribute, and use these materials in any way, even commercially, with proper attribution. For these reasons, we cannot publish previously copyrighted maps or satellite images created using proprietary data, such as Google software (Google Maps, Street View, and Earth). For more information, see our copyright guidelines: http://journals.plos.org/plosone/s/licenses-and-copyright.

Natural Earth (public domain): http://www.naturalearthdata.com/.

Reviewers' comments:

Reviewer's Responses to Questions

**Comments to the Author**

1. Is the manuscript technically sound, and do the data support the conclusions?

Reviewer #1: Partly

Reviewer #2: Yes

2. Has the statistical analysis been performed appropriately and rigorously? 

Reviewer #1: Yes

Reviewer #2: Yes

3. Have the authors made all data underlying the findings in their manuscript fully available?

Reviewer #1: No

Reviewer #2: No

4. Is the manuscript presented in an intelligible fashion and written in standard English?

Reviewer #1: Yes

Reviewer #2: Yes

5. Review Comments to the Author

Reviewer #1: The main claims of the paper and their significance for the discipline

The paper compares several measures of the functional composition of fish communities between two coral reef sites. The two sites differ in habitat “degradation”. The premise of the paper is that, at larger spatial domains, fish communities are functionally similar across sites that differ in habitat metrics. The paper is set up as a test of whether this pattern is preserved at a smaller spatial domain.

I think the authors characterization of the literature is fair and, as such, the objective of the paper is worthwhile. The contribution of the paper to the discipline would be much enhanced by a theory-based explanation of WHY the large-scale pattern should be observed at a finer spatial extent.

The conceptual underpinning is the difference between habitat-specialists and habitat-generalists and the prediction that generalists will fare better when habitat is lost/degraded. I think the authors could do a better job of operationalizing the various verbal concepts they invoke. They discuss “habitat”, “niche”, “degradation”, “functional traits”, and other related terms without clear definitions of each, and how they are operationalized. I am not being frivolous, there is a substantive literature surrounding each of these terms (see for example Hall et al 1997 and Morrison and Hall 2002 for essays about the definitions of “habitat”) and, at minimum, the authors could benefit from stating (1) whose definition they are using and (2) how that is translated to the variables they measure.

Do the data and analyses fully support the claims? If not, what other evidence is required?

I think there are some weaknesses to the study that diminish support in the authors claims. I summarize my suggestions for improvement below, in roughly descending order of importance.

i. No replication of treatment

The authors compare two sites, one degraded and one not. The treatment is, therefore, not replicated and there is very limited basis for assigning differences between the two sites to “degradation”. This is basic study design and ought to be acknowledged clearly.

A related, by less important comment. The main difference in coral composition between the two sites appeared to be due to one species (A. palmata). The authors could comment on how this affects the generality of their study.

ii. Confounding of degradation and tourist visitation

The authors state that the sites differ in two ways: Bonanza = degraded habitat and open to tourists, Linomes = “pristine” habitat and closed to tourists. In addition to the lack of replication, any effect of degradation is confounded with effects of tourism. Tourists have various impacts ion coral refs, e.g. boat anchoring (e.g. Flynn & Forrester 2019) and diving (e.g. Giglio et al 2019) so this is not a hypothetical confounding.

iii. Functional traits could be better-related to “habitat” and its degradation, and are also likely to be strongly affected by the census technique

Fish functional traits could be better justified as relating to habitat or resource use. Diet is clearly relevant, but other variables give the impression of being measured for convenience rather than because of their actual value as indices that are functionally related to habitat use. The functional characterization of the community is also limited and skewed by the method.

The visual census method used is highly biased towards counting diurnal, medium-sized, mid-water fishes and the counting biases should be acknowledged when interpreting the results.

Specifically:-

a) Period of activity. The fish counts were made in the daytime, so it is hard to justify using this variable. Diurnal species were 83% of the total, which is to be expected simply because counts were made during the daylight when most nocturnal species are not reliable countable.

b) Mobility: Most species were classified as “mobile”, but absolute home range figures were not provided. If species classified as “mobile” are not actually resident at the two sites, it is probably inappropriate to match site features (habitat) to counts of their “abundance”.

c) Body size. Medium-sized species were 40% of the total. This may be simply because they are most amenable to counting using the visual transect method. Small fish are hard to see, and big fish are often skittish (scared if divers)

Statistics, and other analyses

The paper is prepared to a high professional standard. It is well-organized, well-written, and the analyses are appropriate.

Figures

A minor comment on the composition of Fig. 1A. The fleshy algal interpolation is clear, but it is difficult to visually judge differences in A. palmata cover. Perhaps consider a different color scheme, larger points, or even a separate paired plot for the A. palmata data.

Data availability

The preliminary forms say there will be some restrictions on data availability. Then, the authors say all data is available with the MS. I could not find a link to the data?

Reviewer #2: 1. Summary of the research and your overall impression

The authors characterize and compare the functional structure of reef fish assemblages at two sites: a highly degraded site and a relatively healthy site. They hypothesized that the relatively healthy site would host a more diverse fish trait composition and exhibit a higher functional diversity compared to the degraded site. They characterized different aspects of functional diversity including functional richness, functional evenness, functional dispersion, and functional originality based on six functional traits. The composition and diversity of fish functional traits was similar between the reefs suggesting that habitat condition does not influence trait composition at small-spatial scales. However, they observed significant differences in functional richness evenness between the reefs. The suggest that the complex microhabitats in the healthy contribute to higher functional richness in the healthy reef, while higher functional evenness may be because of the reduced abundance of fish trait combinations in the degraded reef. They argue that functional redundancy in degraded coral reefs may act insures against habitat disturbances in the short-term. The authors conclude that widespread degradation can lead to functional homogenization of fish communities even at local scales.

2. Overall impression of the manuscript

I wish to congratulate the authors on a very well hypothesized and researched study.

It was enjoyable to read being clear, concise, and easy to follow. The methods are clearly described and repeatable. The data is also well analyzed and presented using relevant statistics and multivariate approaches. The findings are well supported by existing and current literature. The discussion is comprehensive and conclusion is well supported by the evidence provided in the results. I therefore have no major concerns and endorse acceptance of the manuscript following minor corrections:

Abstract

L37 interchange the order from broad to specific as follows: type of traits, variety and range

Results

L43 The phrase…decrease in biodiversity is not ‘hitting the nail on the head’)…suggested edit….Habitat

loss and degradation are leading drivers of declining biodiversity and alteration of ecosystem processes and

functioning….

L268 Check use of the term ‘Fish body size’ vs ‘body size’…’Fish body size’ is mentioned for the first time here, you

can do this on the first mention in L179

Discussion

L359 I find the term 'reef condition' to be broad. I suggest being specific to 'habitat condition'. Then

adjust the next sentence. Despite notable differences in hard coral cover, algae cover, habitat complexity

as well as the number of species, density.....

L368 Delete ‘because’…it is redundant

L458 - 463 Consider breaking this sentence into two smaller sentences. It is a bit long and windy

6. PLOS authors have the option to publish the peer review history of their article (what does this mean?). If published, this will include your full peer review and any attached files.

Reviewer #1: No

Reviewer #2: No

---

## [Author Response · Author response to Decision Letter 0]

11 Oct 2023

R: Done.

R: Done.

“ProgramUNAM-DGAPA-PAPIIT, project IN-205614), granted to L.A.-E.”

R: Done, see details in the following comment. 

“We thank Alba Gonzalez-Posada, Tomás López-Londoño, Kelly Gómez-Campo, Fernando 511 Negrete-Soto and Cecilia Barradas-Ortiz for their help in conducting the fieldwork. The present 512 study was supported by Consejo Nacional de Ciencia y Tecnología (CONACYT) doctoral 513 fellowship (CVU 707432) granted to M.O.-G. This study was funded by Universidad Nacional 514 Autónoma de México (ProgramUNAM-DGAPA-PAPIIT, project IN-205614), granted to L.A.-E”

We note that you have provided funding information that is not currently declared in your Funding Statement. However, funding information should not appear in the Acknowledgments section or other areas of your manuscript. We will only publish funding information present in the Funding Statement section of the online submission form. Please remove any funding-related text from the manuscript and let us know how you would like to update your Funding Statement. Currently, your Funding Statement reads as follows: “ProgramUNAM-DGAPA-PAPIIT, project IN-205614), granted to L.A.-E.” Please include your amended statements within your cover letter; we will change the online submission form on your behalf.

R: We removed the funding-related text from the acknowledgments section, and we now include the funding information to the funding statement.

Upon re-submitting your revised manuscript, please upload your study’s minimal underlying data set as either Supporting Information files or to a stable, public repository and include the relevant URLs, DOIs, or accession numbers within your revised cover letter. For a list of acceptable repositories, please see http://journals.plos.org/plosone/s/data-availability#loc-recommended-repositories.

Any potentially identifying patient information must be fully anonymized.

Important: If there are ethical or legal restrictions to sharing your data publicly, please explain these restrictions in detail. Please see our guidelines for more information on what we consider unacceptable restrictions to publicly sharing data: http://journals.plos.org/plosone/s/data-availability#loc-unacceptable-data-access-restrictions. Note that it is not acceptable for the authors to be the sole named individuals responsible for ensuring data access. We will update your Data Availability statement to reflect the information you provide in your cover letter.

R: we now include the data used to perform the analyses of the manuscript in the supplementary material S3 Table in excel format.

7. We note that [Figure 1] in your submission contain [map/satellite] images which may be copyrighted. All PLOS content is published under the Creative Commons Attribution License (CC BY 4.0), which means that the manuscript, images, and Supporting Information files will be freely available online, and any third party is permitted to access, download, copy, distribute, and use these materials in any way, even commercially, with proper attribution. For these reasons, we cannot publish previously copyrighted maps or satellite images created using proprietary data, such as Google software (Google Maps, Street View, and Earth). For more information, see our copyright guidelines: http://journals.plos.org/plosone/s/licenses-and-copyright.

“I request permission for the open-access journal PLOS ONE to publish XXX under the Creative Commons Attribution License (CCAL) CC BY 4.0 (http://creativecommons.org/licenses/by/4.0/). Please be aware that this license allows unrestricted use and distribution, even commercially, by third parties. Please reply and provide explicit written permission to publish XXX under a CC BY license and complete the attached form.” Please upload the completed Content Permission Form or other proof of granted permissions as an ""Other"" file with your submission. In the figure caption of the copyrighted figure, please include the following text: “Reprinted from [ref] under a CC BY license, with permission from [name of publisher], original copyright [original copyright year].”

Natural Earth (public domain): http://www.naturalearthdata.com/.

R: Figure 1 does not contain third parties’ images. Figure 1A was made from spatial mapping using manta tows to identify and characterize live patches of Acropora palmata on Bonanza and Limones reefs (<5 m). One or two observers were towed behind a small boat holding a GPS (inside a thin plastic cage). Resulting GPS-tracks were used to define the limits of each reef (>5 m depth). Spatial representation and calculation were carried out in ArcGIS 10.3. We now include this description in the methods section in the lines 155-157.

Reviewers' comments:

Reviewer's Responses to Questions

Comments to the Author

1. Is the manuscript technically sound, and do the data support the conclusions?

Reviewer #1: Partly

Reviewer #2: Yes

2. Has the statistical analysis been performed appropriately and rigorously?

Reviewer #1: Yes

Reviewer #2: Yes

3. Have the authors made all data underlying the findings in their manuscript fully available?

Reviewer #1: No

Reviewer #2: No

4. Is the manuscript presented in an intelligible fashion and written in standard English?

Reviewer #1: Yes

Reviewer #2: Yes

5. Review Comments to the Author

Reviewer #1: The main claims of the paper and their significance for the discipline

The paper compares several measures of the functional composition of fish communities between two coral reef sites. The two sites differ in habitat “degradation”. The premise of the paper is that, at larger spatial domains, fish communities are functionally similar across sites that differ in habitat metrics. The paper is set up as a test of whether this pattern is preserved at a smaller spatial domain.

I think the authors characterization of the literature is fair and, as such, the objective of the paper is worthwhile. The contribution of the paper to the discipline would be much enhanced by a theory-based explanation of WHY the large-scale pattern should be observed at a finer spatial extent.

R: We appreciate the reviewer comment, we understand your point and have reworded our text to clarify why we expected consistent patterns of reef fish trait diversity at large-scale and smaller-scale. This is because human activities are increasingly degrading habitats, generalist species are "replacing" specialist species, and the changes in the community structure lead to biotic homogenization and favors an increase in functional similarity of fish assemblages. 

This is particularly relevant to our study because the widespread habitat degradation in the Caribbean region could be favoring the relative increase of functional similarity and simplification of fish communities at smaller scales due to species turnover among previously differentiated fish assemblages. 

We have reworded lines 80-83 and 87-91 to clarify our rationale.

L80-83: “… For instance, human activities are increasingly degrading habitats, leading to the loss of specialist species and favoring the increasing prevalence of generalist species [6,23]. Habitat degradation has also caused a biotic homogenization through changes in community structure and an increase in the taxonomic and functional similarity of reef fish communities over time [e.g., 24].” 

L87-91: “… This is particularly relevant in the Caribbean because the widespread habitat degradation across the region could be favoring the relative increase of functional similarity and simplification of fish communities at smaller scales due to species turnover among previously differentiated fish assemblages.”

The conceptual underpinning is the difference between habitat-specialists and habitat-generalists and the prediction that generalists will fare better when habitat is lost/degraded. I think the authors could do a better job of operationalizing the various verbal concepts they invoke. They discuss “habitat”, “niche”, “degradation”, “functional traits”, and other related terms without clear definitions of each, and how they are operationalized. I am not being frivolous, there is a substantive literature surrounding each of these terms (see for example Hall et al 1997 and Morrison and Hall 2002 for essays about the definitions of “habitat”) and, at minimum, the authors could benefit from stating (1) whose definition they are using and (2) how that is translated to the variables they measure. 

R: We fully agree with the reviewer and appreciate the comment. We now adopt the term habitat quality defined by Hall et al. [1] as “…the ability of the environment to provide conditions appropriate for individual and population persistence…” in L47-48 of the manuscript, and the term habitat degradation such “…as the deterioration of habitat quality…” (according to Pardini et al. [2]) in the L50-51. In addition, we define the term niche according to [3] as “…all that species requires to its population viability in a given environment, as well as including its impacts on that environment…” in L100-101.

Do the data and analyses fully support the claims? If not, what other evidence is required? I think there are some weaknesses to the study that diminish support in the authors claims. I summarize my suggestions for improvement below, in roughly descending order of importance.

i. No replication of treatment

The authors compare two sites, one degraded and one not. The treatment is, therefore, not replicated and there is very limited basis for assigning differences between the two sites to “degradation”. This is basic study design and ought to be acknowledged clearly.

R: We chose to keep the study as simple as possible because these two reefs provide an excellent opportunity to test for the effects of habitat degradation. Please also note we did a considerable number of transects within each reef to provide a great degree of spatial representation of each reef. Below we expand on our arguments. 

In this study we set out to compare two reefs that are under a unique set of circumstances. First, the two reefs are very close to each other, occur under very similar environmental conditions, have a similar reef geomorphology, and until a few decades ago, both had very similar coral communities (see lines 115-121). This allows to contrast the effect of degradation while controlling much of other sources of variation. We did not consider including other sites in this study, as this would result in the inclusion of multiple confounding factors. Although we agree we could account for this source of uncertainty within statistical models, the simplicity of this natural experiment would be overshadowed by complex statistical procedures. We could neither replicate this exact setting (degraded vs ‘healthy’ reef) with other reefs (e.g., three in good condition vs three in poor condition) as we are not aware of reefs similar to Limones elsewhere in our study region. Please note, that Limones reef (in good condition) is probably the last standing reef of its type in the Mesoamerican reef [4]. 

Please also note that in our study we defined the Bonanza and Limones reef communities as the result of multiple censuses sampled randomly at each site, consistent with the scale at which fish communities can respond to fine-scale measured benthic habitat components. Thus, we calculate the ecological and functional metrics at the transect level to have a significant replication and a finer scale when comparing the response of fish communities with their habitat quality on both reefs. 

We now further clarify this statement in the methods section. Lines 173-176 and 226-228.

L173-176: “To survey the reef fish communities, underwater visual censuses were haphazardly performed on Bonanza and Limones. We defined the Bonanza and Limones reef communities as the result of multiple censuses sampled randomly at each site, consistent with the scale at which fish communities can respond to fine-scale measured benthic habitat components.”

L226-228: “We calculate the ecological and functional metrics at the transect level to have a significant replication and a finer scale when comparing the response of fish communities with their habitat quality on both reefs.”

A related, by less important comment. The main difference in coral composition between the two sites appeared to be due to one species (A. palmata). The authors could comment on how this affects the generality of their study.

R: Yes, the reviewer is correct. A. palmata is well-known as the major reef-building coral in shallow-water Caribbean reefs [4,5]. Thus, the main difference in coral composition between our sites is due to the presence of large-stands of live A. palmata in Limones, while Bonanza is represented largely by large stands of dead A. palmata in different degrees of degradation. We now highlight this explanation in the methods section lines 118-121: “It is well-known that Acropora palmata is the most structurally complex in the Caribbean and previous studies have identified Limones as the reef site with the highest structural complexity and physical functionality among the northern Caribbean coral reefs[35,36]. Bonanza has been described as a heavily degraded reef at least since the mid-2000s [37].”

ii. Confounding of degradation and tourist visitation

The authors state that the sites differ in two ways: Bonanza = degraded habitat and open to tourists, Linomes = “pristine” habitat and closed to tourists. In addition to the lack of replication, any effect of degradation is confounded with effects of tourism. Tourists have various impacts ion coral refs, e.g. boat anchoring (e.g. Flynn & Forrester 2019) and diving (e.g. Giglio et al 2019) so this is not a hypothetical confounding.

R: Thank you for pointing this out. We were not clear enough in the description of our study sites. Bellow we provide details to support that the effect of tourism is, in this case, unrelated to the condition of Bonanza. 

First, both reefs are protected within the Puerto Morelos Reef National Park since 1998, the visitation is regulated and enforced by the Marine Park [4]. In Bonanza, visitation is restricted to a small portion of the reef (~5% of the reef area), where a mooring buoy is provided (anchoring is prohibited). Given that Bonanza is a shallow reef diving is not feasible/permitted, and only snorkeling while using a life vest is allowed. 

More importantly, visitation was permitted in both reefs until recently, when Bonanza was already in a degraded state [4]. It was not until 2014 that Limones was decreed a critical habitat for the conservation of A. palmata. Before that time, tourism visitation was allowed in this reef. Bonanza has been described as a heavily degraded reef at least since the mid-2000s [6]. 

We have reworded our description of the study area in the lines 126-129.

L126-129: “…Both reefs are protected as part of the Puerto Morelos Reef National Park since 1998 and were open to visitation until 2014, when Limones was decreed a critical habitat for the conservation of A. palmata. Since then, Limones has been closed to visitation, but as Bonanza was already in a degraded state, it has remained open for visitation [36].”

iii. Functional traits could be better-related to “habitat” and its degradation, and are also likely to be strongly affected by the census technique

Fish functional traits could be better justified as relating to habitat or resource use. Diet is clearly relevant, but other variables give the impression of being measured for convenience rather than because of their actual value as indices that are functionally related to habitat use. The functional characterization of the community is also limited and skewed by the method.

R: Thank you for this comment. All traits used in this study were carefully considered and chosen before conducting the analyses.

In Table S1 we provide the rationale of why we chose each trait, based on their ecological importance and their response/effect approach against habitat changes. Please also note that we now provide some examples of how the functional traits are related with the habitat and resource use in lines 200-203. In the following comments we respond to your concerns regarding each trait. 

The visual census method used is highly biased towards counting diurnal, medium-sized, mid-water fishes and the counting biases should be acknowledged when interpreting the results.

R: We agree with the reviewer. However, please note that to reduce the observer bias to the minimum, all visual censuses were conducted by trained scientific observers that have jointly carried out similar surveys in the Caribbean for several years (see methods section lines 186-188). Also, we added some lines in the Discussion acknowledging the limitations of the visual censuses (see lines 422-426): “…The current study focused mainly on conspicuous reef fishes in shallow coral reefs, where large and highly mobile fish species are generally not abundant as they move to deeper areas. Also, underwater visual censuses underestimate the number and density of cryptobenthic species, resulting in a scarcity of quantitative data on this assemblage, both regionally and globally [57], making it difficult to include them in trait-based approaches.”

Specifically:-

a) Period of activity. The fish counts were made in the daytime, so it is hard to justify using this variable. Diurnal species were 83% of the total, which is to be expected simply because counts were made during the daylight when most nocturnal species are not reliable countable.

R: Thank you for this comment, as it gave us the opportunity to expand on the rationale to use this particular trait. We consider highlighting the importance of the period of activity trait (i.e., foraging activity) not only in diurnal species but also in crepuscular (i.e., diurnal-nocturnal) and nocturnal species, since some studies have found correlations between diel activity patterns and trophic level, habitat use and depth range (e.g., [7,8]). Particularly, the level of habitat quality could directly affect the habitat occupancy, species interactions, and consequently, the diel activity patterns of reef fishes. For example, most nocturnal species reduce their risk of predation by finding shelter in preserved habitat during the daytime, and these species interactions may change if the habitat is in a degraded state. We add this explanation to Table S1 of the supplementary material of the manuscript. 

b) Mobility: Most species were classified as “mobile”, but absolute home range figures were not provided. If species classified as “mobile” are not actually resident at the two sites, it is probably inappropriate to match site features (habitat) to counts of their “abundance”.

R: Thank you for this comment, The home range for each species was qualitatively categorized into one of three levels: sedentary-territorial species (species move less than a few 10 m2), mobile within-reef (species that can move 10s of meters within the same reef), and very mobile among reefs (species that can move distances over 100 m2 and usually move between habitats/reefs). Therefore, this trait does represent a mobility gradient to represent species with different levels of attachment to reef structures or habitats. Furthermore, most species classified as “mobile” are still resident of relatively small areas and don’t tend to move between different reefs (across kms). We now added some lines to better describe and categorize this functional trait in Table 1 of the manuscript. Furthermore, it is worth mentioning that this functional trait has been successfully used in previous studies to describe the functional structure of reef fish assemblages at different scales [9,10]. 

c) Body size. Medium-sized species were 40% of the total. This may be simply because they are most amenable to counting using the visual transect method. Small fish are hard to see, and big fish are often skittish (scared if divers)

R: We agree that underwater visual censuses underestimate large highly mobile species and cryptobenthic species. Our study focused mainly on conspicuous reef fishes in shallow coral reefs, where large and highly mobile fish species are generally not abundant as they move to deeper areas. In addition, although a large proportion of diversity is composed of cryptobenthic reef fishes, the difficulty of accurately assessing and sampling these species has resulted in a poor body of quantitative data on cryptobenthic assemblages, both regionally and globally [11] making it difficult to also include them in trait-based approaches. We have added the following to lines 422-423.

L422-423: “The current study focused mainly on conspicuous reef fishes in shallow coral reefs, where large and highly mobile fish species are generally not abundant, as they tend to move to deeper areas…”

Statistics, and other analyses

The paper is prepared to a high professional standard. It is well-organized, well-written, and the analyses are appropriate.

R: We thank the reviewer for his comment.

Figures

A minor comment on the composition of Fig. 1A. The fleshy algal interpolation is clear, but it is difficult to visually judge differences in A. palmata cover. Perhaps consider a different color scheme, larger points, or even a separate paired plot for the A. palmata data.

R: we corrected the figure 1A with a different color palette and larger points.

Data availability

The preliminary forms say there will be some restrictions on data availability. Then, the authors say all data is available with the MS. I could not find a link to the data?

R: We now provide the data information used to perform the ecological and functional analyses in the supplementary material S3 Table in excel format.

Reviewer #2: 1. Summary of the research and your overall impression

The authors characterize and compare the functional structure of reef fish assemblages at two sites: a highly degraded site and a relatively healthy site. They hypothesized that the relatively healthy site would host a more diverse fish trait composition and exhibit a higher functional diversity compared to the degraded site. They characterized different aspects of functional diversity including functional richness, functional evenness, functional dispersion, and functional originality based on six functional traits. The composition and diversity of fish functional traits was similar between the reefs suggesting that habitat condition does not influence trait composition at small-spatial scales. However, they observed significant differences in functional richness evenness between the reefs. The suggest that the complex microhabitats in the healthy contribute to higher functional richness in the healthy reef, while higher functional evenness may be because of the reduced abundance of fish trait combinations in the degraded reef. They argue that functional redundancy in degraded coral reefs may act insures against habitat disturbances in the short-term. The authors conclude that widespread degradation can lead to functional homogenization of fish communities even at local scales.

R: We thank the reviewer for his/her comment.

2. Overall impression of the manuscript

I wish to congratulate the authors on a very well hypothesized and researched study. It was enjoyable to read being clear, concise, and easy to follow. The methods are clearly described and repeatable. The data is also well analyzed and presented using relevant statistics and multivariate approaches. The findings are well supported by existing and current literature. The discussion is comprehensive, and conclusion is well supported by the evidence provided in the results. I therefore have no major concerns and endorse acceptance of the manuscript following minor corrections: 

R: We appreciate the reviewer for the comment.

Abstract

L37 interchange the order from broad to specific as follows: type of traits, variety and range. 

R: Done.

Results

L43 The phrase…decrease in biodiversity is not ‘hitting the nail on the head’)…suggested edit….Habitat loss and degradation are leading drivers of declining biodiversity and alteration of ecosystem processes and functioning….

R: Done.

L268 Check use of the term ‘Fish body size’ vs ‘body size’…’Fish body size’ is mentioned for the first time here, you can do this on the first mention in L179. 

R: we corrected and standardized the use of the term ‘fish body size’ in the manuscript.

Discussion

L359 I find the term 'reef condition' to be broad. I suggest being specific to 'habitat condition'. Then adjust the next sentence. Despite notable differences in hard coral cover, algae cover, habitat complexity as well as the number of species, density.....

R: Done. 

L368 Delete ‘because’…it is redundant 

R: Done.

L458 - 463 Consider breaking this sentence into two smaller sentences. It is a bit long and windy.

R: We modified the sentence to give more clarity to the text. L484-487.

L484-487: Thus, functional richness, which is strongly influenced by the number of species, reflected a higher trait diversity in Limones, whereas functional evenness, which is rather influenced by the density of individuals, showed a more regular distribution of trait combinations in Bonanza [13,14].

6. PLOS authors have the option to publish the peer review history of their article (what does this mean?). If published, this will include your full peer review and any attached files.

Do you want your identity to be public for this peer review? For information about this choice, including consent withdrawal, please see our Privacy Policy.

Reviewer #1: No

Reviewer #2: No

References

1. Hall LS, Krausman PR, Morrison ML, Hall LS, Krausman PR, Morrison ML. The Habitat Concept and a Plea for Standard Terminology. Wildl Soc Bull. 1997;25: 173–182. 

2. Pardini R, Nichols E, Püttker T. Biodiversity response to habitat loss and fragmentation. Encycl Anthr. 2017;1–5: 229–239. doi:10.1016/B978-0-12-809665-9.09824-4

3. Clavel J, Julliard R, Devictor V. Worldwide decline of specialist species: Toward a global functional homogenization? Front Ecol Environ. 2011;9: 222–228. doi:10.1890/080216

4. Rodríguez-Martínez RE, Banaszak AT, McField MD, Beltrán-Torres AU, Álvarez-Filip L. Assessment of Acropora palmata in the Mesoamerican reef system. PLoS ONE. 2014;9: 1–7. doi:10.1371/journal.pone.0096140

5. Cramer KL, Jackson JBC, Donovan MK, Greenstein BJ, Korpanty CA, Cook GM, et al. Widespread loss of Caribbean acroporid corals was underway before coral bleaching and disease outbreaks. Sci Adv. 2020;6. doi:10.1126/sciadv.aax9395

6. Jordan Dahlgren E. Atlas de los arrecifes coralinos del Caribe Mexicano. pt. 1: El sistema continental. 1993. 

7. Arndt E, Evans J. Diel activity of littoral and epipelagic teleost fishes in the Mediterranean Sea. Rev Fish Biol Fish. 2022;32: 497–519. doi:10.1007/s11160-022-09697-9

8. Campanella F, Auster PJ, Christopher Taylor J, Muñoz RC. Dynamics of predator-prey habitat use and behavioral interactions over diel periods at sub-tropical reefs. PLoS ONE. 2019;14: 1–22. doi:10.1371/journal.pone.0211886

9. Mouillot D, Villéger S, Parravicini V, Kulbicki M, Arias-González JE, Bender M, et al. Functional over-redundancy and high functional vulnerability in global fish faunas on tropical reefs. Proc Natl Acad Sci U S A. 2014;111: 13757–13762. doi:10.1073/pnas.1317625111

10. Quimbayo JP, Silva FC, Mendes TC, Ferrari DS, Danielski SL, Bender MG, et al. Life-history traits, geographical range, and conservation aspects of reef fishes from the Atlantic and Eastern Pacific. Ecology. 2021;102: 4455016. doi:10.1002/ecy.3298

11. Brandl SJ, Goatley CHR, Bellwood DR, Tornabene L. The hidden half: ecology and evolution of cryptobenthic fishes on coral reefs. Biol Rev. 2018;93: 1846–1873. doi:10.1111/brv.12423

---

## [Decision Letter · Decision Letter 1]

20 Nov 2023

Similar functional composition of fish assemblages despite contrasting levels of habitat degradation on shallow Caribbean coral reefs

PONE-D-23-03979R1

Dear Dr. Alvarez-Filip,

We’re pleased to inform you that your manuscript has been judged scientifically suitable for publication and will be formally accepted for publication once it meets all outstanding technical requirements.

Kind regards,

Frank O. Masese, Ph.D

Academic Editor

PLOS ONE

Additional Editor Comments (optional):

Reviewers' comments:

Reviewer's Responses to Questions

**Comments to the Author**

1. If the authors have adequately addressed your comments raised in a previous round of review and you feel that this manuscript is now acceptable for publication, you may indicate that here to bypass the “Comments to the Author” section, enter your conflict of interest statement in the “Confidential to Editor” section, and submit your "Accept" recommendation.

Reviewer #2: All comments have been addressed

2. Is the manuscript technically sound, and do the data support the conclusions?

Reviewer #2: Yes

3. Has the statistical analysis been performed appropriately and rigorously? 

Reviewer #2: Yes

4. Have the authors made all data underlying the findings in their manuscript fully available?

Reviewer #2: Yes

5. Is the manuscript presented in an intelligible fashion and written in standard English?

Reviewer #2: Yes

6. Review Comments to the Author

Reviewer #2: The authors have addressed review comments sufficiently. I find the paper is now in an acceptable state for publication

7. PLOS authors have the option to publish the peer review history of their article (what does this mean?). If published, this will include your full peer review and any attached files.

Reviewer #2: No

---

## [Editor Report · Acceptance letter]

14 Dec 2023

PONE-D-23-03979R1 

PLOS ONE

Dear Dr. Alvarez-Filip, 

I'm pleased to inform you that your manuscript has been deemed suitable for publication in PLOS ONE. Congratulations! Your manuscript is now being handed over to our production team.

Kind regards, 

on behalf of

Dr. Frank O. Masese 

Academic Editor

PLOS ONE